# lncRNA Biomarkers of Glioblastoma Multiforme [note 1]

**DOI:** 10.3390/biomedicines12050932

**Published:** 2024-04-23

**Authors:** Markéta Pokorná, Marie Černá, Stergios Boussios, Saak V. Ovsepian, Valerie Bríd O’Leary

**Affiliations:** 1Department of Medical Genetics, Third Faculty of Medicine, Charles University, Ruská 87, Vinohrady, 10000 Prague, Czech Republic; marie.cerna@lf3.cuni.cz (M.Č.); valerie.oleary@lf3.cuni.cz (V.B.O.); 2Department of Medical Oncology, Medway NHS Foundation Trust, Gillingham ME7 5NY, UK; stergios.boussios@nhs.net; 3Faculty of Medicine, Health, and Social Care, Canterbury Christ Church University, Canterbury CT2 7PB, UK; 4Faculty of Life Sciences & Medicine, School of Cancer & Pharmaceutical Sciences, King’s College London, Strand, London WC2R 2LS, UK; 5Kent Medway Medical School, University of Kent, Canterbury CT2 7LX, UK; 6AELIA Organization, 9th Km Thessaloniki-Thermi, 57001 Thessaloniki, Greece; 7Faculty of Engineering and Science, University of Greenwich London, Chatham Maritime, Kent ME4 4TB, UK; s.v.ovsepian@greenwich.ac.uk; 8Faculty of Medicine, Tbilisi State University, Tbilisi 0177, Georgia

**Keywords:** lncRNA, noncoding RNA, glioblastoma multiforme, glioma, plasma, serum, blood, biomarker, liquid biopsy, AC016405.3, ADAMTs9-AS2, AGAP2-AS1, AHIF, ANRIL, lncRNA-ATB, CASC2, CASC7, CASC9, CCND2-AS1, CRNDE, DCST1-AS1, DGCR5, DLEU1-AS1, ECONEXIN, LINC00461, FAM66C, GAS5, H19, HMMR-AS1, HOTAIR, HOTAIRM1, HOXA-AS2, HOXB13-1, HOTTIP, HULC, KTN1-AS1, LINC00467, LINC00565, LINC00641, LINC01393, LINC01426, LINC01446, LINC01494, LINC01503, LINC01711, LINC02283, LINC-ROR, lnc-TALC, MAFG-DT, MALAT1, MATN1-AS1, MDC1-AS, MEG3, MIAT, MIR210HG, MNX1-AS1, NCK1-AS1, NEAT1, PART1, PARTICLE, PCAT1, PCA1, PVT1, RBPMS-AS1, RPSAP52, RUNX1-IT1, SAMMSON, SOX2-OT, TALNEC2, TP73-AS1, TSLC1-AS1, TUSC7, TUG1, TUNAR, UCA1, XIST, ZEB1-AS1, ZBED3-AS1

## Abstract

Long noncoding RNAs (lncRNAs) are RNA molecules of 200 nucleotides or more in length that are not translated into proteins. Their expression is tissue-specific, with the vast majority involved in the regulation of cellular processes and functions. Many human diseases, including cancer, have been shown to be associated with deregulated lncRNAs, rendering them potential therapeutic targets and biomarkers for differential diagnosis. The expression of lncRNAs in the nervous system varies in different cell types, implicated in mechanisms of neurons and glia, with effects on the development and functioning of the brain. Reports have also shown a link between changes in lncRNA molecules and the etiopathogenesis of brain neoplasia, including glioblastoma multiforme (GBM). GBM is an aggressive variant of brain cancer with an unfavourable prognosis and a median survival of 14–16 months. It is considered a brain-specific disease with the highly invasive malignant cells spreading throughout the neural tissue, impeding the complete resection, and leading to post-surgery recurrences, which are the prime cause of mortality. The early diagnosis of GBM could improve the treatment and extend survival, with the lncRNA profiling of biological fluids promising the detection of neoplastic changes at their initial stages and more effective therapeutic interventions. This review presents a systematic overview of GBM-associated deregulation of lncRNAs with a focus on lncRNA fingerprints in patients’ blood.

## 1. Introduction

To date, more than 100,000 transcripts that are not translated and do not encode proteins have been identified [1] (see Figure 1). These noncoding molecules are highly heterogeneous and vary in length, function, location in genome, and distribution in various cells or tissue types. Noncoding RNAs (ncRNAs) form a large heterogeneous set of functional RNA molecules that are transcribed from different locations throughout the genome. Although ncRNAs are not translated into proteins, they play an important role in physiological processes and in the regulation of gene expression. The importance of the noncoding transcriptome is supported by the direct correlation between the proportion of ncRNAs in the genome of organisms and their developmental complexity [2]. There is no such correlation in the number of protein-coding genes [3]. Reports also show that the number of lncRNA types in neural tissue correlates with the complexity of the nervous system. The human brain has been found to have the highest number of lncRNA types in all organisms studied to date [4,5].

Transcripts longer than 200 nucleotides belong to a large group of long noncoding RNAs (lncRNAs) [2,7]. Their total number is increasing due to more sensitive detection methods and is greater than the sum of all protein-coding genes [5]. lncRNAs are mainly transcribed by polymerase II and may subsequently undergo post-transcriptional modifications. Through interaction with proteins and regulatory segments of the genome, lncRNAs of neurons are involved in the control of many cellular processes including differentiation, proliferation, migration, and signalling, as well as in an array of epigenetic mechanisms [1,7]. lncRNAs have been detected in the nucleus, nucleolus, cytoplasm, and mitochondria [1]. There is rising evidence suggesting a mechanistic link between many human diseases, including cancer, and lncRNA dysregulations, making lncRNA molecules potential therapeutic targets and biomarkers for diseases, which may facilitate the detection and diagnosis of various disorders and diseases [1].

The exact function of most lncRNAs remains unknown, with substantial evidence suggesting that their localisation can predict likely role they play in the cell. Indeed, the transcripts that prevail in the nucleus are involved in the regulation of gene expression, chromatin modification, and imprinting [8]. lncRNAs prevalent in the cytoplasm, on the other hand, are involved in mRNA splicing and the regulation of protein translation and may also be precursors for small noncoding RNAs (sncRNAs), e.g., microRNAs (miRNAs) [1,8].

Several lncRNA-specific databases have been created that contain information on their origin, functions, and action mechanisms (e.g., LNCipedia 5.2; lncRNAfunc), along with their alternative names and various identifiers, e.g., gene ID, Hugo nomenclature, and Ensembl tags for both genes and transcripts. Most databases do not list all the data and all the names for a given lncRNA, and some use their specific lncRNA identification system. This makes it difficult to find information about a particular lncRNA molecule not only in databases but also in peer-reviewed publications. Hence, there is pressing need for the use of uniform and standardized lncRNA nomenclature to improve communication and avoid confusion or the duplication of individual molecules. Due to the constant new discoveries related to lncRNAs, in many cases, the same molecule with different functions is identified as different lncRNAs [9,10]. There is a lack of consensus and a need for standardized nomenclature to avoid duplications and confusion in the field. Some lncRNAs, for instance, can appear in search results under two or more names and can be easily taken as multiple distinct lncRNAs.

The response of lncRNAs to glioblastoma multiforme (GBM), with emerging recognition of their detection and prognostic relevance, makes their profiling and analysis of prime relevance to the diagnosis and therapy of this malignant brain condition. GBM refers to the most common and aggressive malignant brain tumour in adults that resists conventional therapy, which includes surgical resection followed by radiation therapy and chemotherapy [10]. GBM is considered a whole brain disease because the neoplastic cells are highly invasive, infiltrating in surrounding tissue and spreading beyond the lesion area. This characteristic makes tumour resection highly challenging and leads to frequent post-surgery recurrences, which are the main cause of mortality [11]. Despite the relatively low incidence (3–4 cases per 100,000 people), GMB remains one of the greatest challenges and priorities for research and clinical translation, owing to its severity and very high mortality. On average, treated patients live 14–16 months from the first diagnosis, with only 5–10% of patients surviving 5 years from the manifestation of the disease [11]. The effectiveness of treatment and progress are largely hampered by the high infiltration of malignant tissue and the heterogeneity of neoplastic cells. In addition to malignant neoplastic cells, the lesions of GBM typically contain endothelial cells, neurones, astrocytes, oligodendrocytes, microglia, and non-cellular components such as apocrine and paracrine signalling factors, exosomes, and other cell types and tissue debris [12]. These components are typically segregated into several distinct compartments known as tumour niches, which may differ morphologically and functionally even within a single tumour. Numerous studies confirm the involvement of lncRNAs in many molecular processes in GBM tissue [13,14,15,16]. Revealing their precise function could aid in the discovery of new therapeutic approaches. These molecules may also serve well as biomarkers—directly in tumour tissue—for more accurate diagnosis and the initiation of more effective therapy after tumour resection.

In this article, we provided a systematic review of lncRNAs associated with GBM, with their response in the disease and diagnostic relevance as biomarkers. Like in several other cancer types, the classification of brain cancers remains challenging, with the term glioma often used also for glioblastoma multiforme cell lines and for tissues from patients with a confirmed diagnosis of GBM. We refer to the commonly used names of lncRNA deregulated in GMB, describe their response to GBM and other cancer types, and discuss their localizations as well as identifiers presented by the Ensembl gene (ENSGs) database.

## 2. Methodology

A systematic review of GBM-associated lncRNAs was generated by searching several databases. First, the PubMed database of the National Library of Medicine was used, where the keywords lncRNA, glioblastoma multiforme, and glioma were entered. From these results, a summary was compiled containing the name of the lncRNA, its role and function in GBM, expressional changes, and comparison with other cancers. In the case of the keyword glioma, it was checked whether it was a GBM cell line, grade IV malignancy in the case of tissues, diagnosis of GBM in patients, or another type of glioma. The given lncRNAs were searched in the databases lncRNAfunc (https://ccsm.uth.edu/lncRNAfunc/, accessed on 31 January 2024), LNCipedia 5.2, and Ensembl (https://www.ensembl.org/index.html, accessed on 31 January 2024), from which additional data—alternative names, gene location, class, and Ensembl gene ID—were added to the list.

## 3. Genome Localisation and Expression of lncRNA

DNA segments from which lncRNAs are transcribed can occur almost anywhere in the human genome. LncRNA molecules can be divided into several groups based on their genome location. Sequences of intron lncRNAs are found in the introns of protein-coding genes. Intergenic lncRNAs (lincRNAs), on the other hand, are in the region between the two coding genes, whereas enhancer lncRNAs (elncRNAs) are localized in the enhancer regions of protein-coding genes. Sequences for lncRNAs, thus, may overlap with the exon, intron, or both parts of a gene, or they may overlap the entire sequence of a protein-coding gene. Importantly, unlike the protein-coding genome, the genome-encoding lncRNA can be localised on both strands of DNA and be transcribed in both directions. Genomic sequences within these transcription units can be shared not only with coding regions but also with each other in both sense and antisense directions [1,17]. In most cases, lncRNA sequences are transcribed by RNA Polymerase II and rarely by RNA Polymerase I or III [18]. The resulting transcripts can be post-transcriptionally modified in a manner shared with protein transcripts, involving the binding of 7-methylguanosine at the 5′ end, polyadenylation at the 3′ end, or splicing [19,20]. Reverse editing can produce circular RNAs (circRNAs), which are another subtype of lncRNA. CircRNAs form covalently closed loops, which makes them resistant to nucleases. Like lncRNA, circRNAs have been detected in biological fluids such as blood, cerebrospinal fluid, and urine, making them potential biomarkers for various disease conditions. Some of these molecules have been associated with different forms of cancers, including gliomas [17].

The expression of lncRNAs is highly specific to various tissue types, with their profiles responding to disease conditions, as well as developmental stage, circadian rhythms, and other variations [20,21]. Quantitative studies suggest that the specificity index of lncRNAs is significantly higher than that of the transcriptome of protein-coding genes. This supports the notion of lncRNAs as regulators of gene expression in specific cell types [4]. Most types of lncRNA have been found in the nervous tissue, which is composed of many cell types that require highly complex regulatory processes. The latter are influenced by lncRNA molecules, which play an important role in the development, maintenance, and influence of neural functionality, contributing to brain mechanisms. The set of lncRNAs in human brain tissue differs from other primate brains by a greater degree than the transcriptome of encoding genes, with the extent of the differences correlating with the developmental stage, functionality, and disease state [4].

## 4. Molecular Mechanisms Underlying lncRNA Functions

To date, studies have implicated lncRNAs in almost all processes of gene expression regulation, including chromosome inactivation, imprinting, chromatin dynamics, protein modification, and nucleic acid stability [22,23]. The expression of lncRNAs can be influenced by a variety of factors, including environment, stress, and the pathophysiological state of the cell. The genes for lncRNAs may be subject to epigenetic modifications, such as promoter methylation [24].

There are four basic molecular mechanisms by which lncRNAs can interact with biomolecules and influence their activity [17]:Signalling, where lncRNAs are transcribed at a specific site and time in a cell type-specific manner, inducing or governing an active signalling event (Figure 2).Decoying, where lncRNAs serve as decoys for target proteins. The lncRNA molecules occupy the binding site, and the proteins cannot interact with DNA (Figure 3a). In this way, lncRNA can interact with transcription factors, repressors, chromatin modifiers, and other proteins. Within this regulatory mode, lncRNAs can also interact with miRNAs (Figure 3b). Specific lncRNAs act as sponges for some miRNAs, i.e., the lncRNA binds to the miRNA, which then cannot perform its function.Guiding, where lncRNA molecules control the placement of ribonucleoprotein complexes at specific target sites, with precision effects (Figure 4).Scaffolding, where transcripts act as scaffolds for other molecules that can bind to a given lncRNA to form a ribonucleoprotein complex (Figure 5).

A large proportion of lncRNAs use more than one of the mechanisms described above to regulate cellular processes and, thus, can perform multiple functions. Therefore, lncRNAs cannot be strictly divided into these four groups [17,25].

## 5. Functions of lncRNAs in Cancer

lncRNA molecules are involved in almost all cellular processes, including growth, development, and differentiation. They also participate in many signalling pathways and mechanisms with p53 signalling, effects of growth hormones, glucose metabolism, cytokine expression, the V(D)J recombination of immune cells, and inflammation [18]. Mutations or the altered expression of lncRNAs have been shown to lead pathophysiological changes, contributing to a variety of cancers [18,26], neurological and neurodegenerative diseases [27,28], and genetic conditions (e.g., phenylketonuria) [29]. Furthermore, guide lncRNAs form complexes with regulatory or enzymatically active proteins, targeting them towards specific gene promoters or genomic loci, thus regulating downstream signalling events and gene expressions [22]. Using genome-wide RNA-Seq analyses, numerous lncRNAs have been identified, exhibiting either upregulation or downregulation in various forms of malignancies, including renal, breast, and brain cancer [30]. Among these lncRNAs, MALAT1, RCAT1, DUXAP9, TCL6, LINC00342, AGAP2 Antisense1, DLEU2, NNT-AS1, LINC00460, and Lnc-LSG1 are, for example, specific to renal cancer, while changes in HOTAIR, ANRIL, ZFAS1, HOTAIRM1, PVT1, MALAT1, and LNP1 are associated with breast and brain cancer [31].

Determining the exact function of a given lncRNA molecule is difficult, as in most cases, changes in their expression do not cause phenotypic alterations. Based on previous studies, some lncRNAs have been assigned as oncogenic (MALAT1, PCA3, HOTAIR, H19, PARTICLE, etc.) or as tumour suppressors (GAS5, MEG3, TERRA, etc.) [17,18,32]. Some lncRNAs may exhibit variability in their properties and effects depending on the type of cancer. For instance, lncRNA AC016405.3 has tumour suppressor function in GBM, while at a higher concentration, it is considered oncogenic for breast cancer [33,34]. lncRNA molecules are specifically expressed in certain types of cancer, with the majority changing also blood or urine. Given the relatively large number of different types of lncRNA and their high tissue specificity, lncRNAs are explored as potential biomarkers for various diseases. A good example is the lncRNA PCA3, whose increased expression signals a prostate cancer and can be detected in urine, together with enhanced MALAT1 and LincRNA-p21 [17]. The expressional changes of some cancer-specific lncRNAs have been shown to correlate with the degree of malignancy, stage of the disease, metastasis, or prognosis [35]. They have also been associated with resistance to therapy and subsequent tumour recurrence [36]. For instance, lncARSR, which showed high expression levels in sunitinib-resistant renal cancer cells, has been found to be essential for the resistant phenotype, through competition with endogenous RNA for miR-34 and miR-449, leading to the upregulation of AXL/c-MET and the activation of STAT3, AKT, and ERK pathways. Remarkably, lncARSR was identified as a predictive marker for poor response in patients with renal cancer, with emerging data suggesting its exosomal release from therapy-resistant cells, thereby conferring treatment resistance [30].

## 6. lncRNAs and GBM

Given the large number of lncRNA types in healthy brains and the extensive heterogeneity of GBM tissue, it is necessary to consider these transcripts when looking into grading the condition, making a prognosis, or exploring their response to experimental therapies. Studies have shown a link between lncRNAs and many processes implicated in the formation and growth of GBM. lncRNA transcripts are involved in cell proliferation (MIAT) [37], cell apoptosis (MALAT1) [38], cell invasion (ATB) [39,40], angiogenesis (HULC) [41], DNA damage response (PCAT1) [42], cell cycle regulation (CASC7) [43], the regulation of the tumour microenvironment (FAM66C) [44], hypoxia (MIR210HG) [45], BBB permeability (TUG1) [46], tumour progression (TUNAR) [47], recurrence (TALC) [48], resistance to temozolomide (TMZ; ADAMTs9-AS2) [49], radiation resistance (HMMR-AS1) [50], and others (see Table 1). lncRNAs can engage directly, through various molecular processes and mechanisms, or indirectly, through the regulation of miRNAs using methylation or by affecting chromatin modification [36]. lncRNAs also may control the microenvironment of GBM, where they can influence the activity of cytokines and growth factors [51]. Finally, lncRNA molecules affect cancer stem cells and, thus, participate in tumorigenesis, recurrence, and resistance to therapy [36]. Accordingly, deregulated levels of lncRNA were detected in resected GBM tissue, and their analysis can provide more accurate differential diagnoses. Expression profiles of different lncRNAs can be also used to determine the grade of glioma and its subtype [13]. Importantly, the dynamics of lncRNAs circulating in the blood can be also used for determining prognosis and monitoring GBM response to treatment.

## 7. lncRNA Biomarker in Diagnostic and Clinical Use

lncRNAs can regulate gene expression by binding to transcription factors and competing for binding sequences for miRNAs, thus inhibiting their action. They can also bind to regulatory proteins and participate in the formation of ribonucleoprotein complexes and induce the modification of chromatin. Finally, lncRNAs can regulate mRNAs at several levels, from translational inhibition and splicing to degradation, thus effecting protein synthesis and function [105]. Changes in the expression of various lncRNAs have been detected in association with many diseases, including cancer [17], depression [28], cardiovascular disease, and others [28,106]. These properties of lncRNAs render them as potential therapeutic targets and instructive biomarkers for difficult-to-diagnose diseases [1,21]. A good example is the clinical application of the lncRNA deregulation of prostate cancer antigen 3 (PCA3). In 2012, the FDA approved a diagnostic test for prostate cancer based on the detection of the elevated expression of lncRNA PCA3 in urine [17].

Reports suggest that some lncRNAs have better diagnostic and prognostic properties than more mainstream and standardised biomarkers [105]. lncRNA molecules meet all major requirements for biomarkers used in clinical diagnostics. They are produced continuously in cells, respond to homeostatic and environmental challenges, are secreted into biological fluids, and can be readily detected via analytical methods. Changes in the levels of these molecules in cells and in biological fluids, therefore, provide valuable information about the alterations to health and disease states [107]. The fact that most lncRNAs are relatively stable, can be released in body fluids including plasma, serum, urine, and cerebrospinal fluid, and can be readily detectable makes them highly suitable as biomarkers [26]. lncRNAs have also been detected in the exosomes of biological fluids, including blood (see Table 2), which makes them better protected from ribonucleases and more stable over time. In this form, lncRNA molecules are also protected from the effects of repeated thawing, assisting in their detection and research [108,109]. Expression levels of lncRNA can be quantified using highly advanced and sensitive laboratory methods such as real-time PCR, NGS, RNA microarrays, and RNA-Seq, which are becoming increasingly available. Importantly, changes in lncRNA levels in tissues and bodily fluids may also reflect alterations in the response of the body to therapeutic intervention [106,110].

For some diseases, it appears that the detection of changes in a single lncRNA is not specific enough to qualify lncRNAs as biomarkers. Indeed, a large percentage of lncRNAs are abnormally expressed in multiple diseases (see Table 2). Rising data show that stress and pathological changes in anatomically related structures, types of tissue, or embryologically interrelated organs are characterised by the abnormal expression of a similar set of lncRNAs [109]. Thus, it is necessary to identify a set of lncRNAs, called a signature or fingerprints, that corresponds to a given disease [109] (see Table 3). An example is the combination of three lncRNAs SPRY4-IT1, ANRIL, and NEAT1. These lncRNAs are abnormally regulated in the blood plasma of patients with non-small cell lung cancer. More than 90% specificity and 80% sensitivity have been achieved in the diagnosis of this disease using the detection of this set of lncRNAs [124]. The analysis of lncRNAs in blood plasma can also be used to determine the prognosis. Changes in the regulation of lncRNAs XLOC_014172 and LOC149086, for example, can distinguish metastatic hepatocellular carcinoma from nonmetastatic carcinoma, with a specificity and sensitivity of more than 90% and an AUC of 0.934 [109].

The detection of changes in the expression of one or more lncRNAs (also as part of liquid biopsy), thus, might be an effective approach for the early diagnosis of various diseases (Figure 6), to ensure more targeted and personalized interventions with better therapeutic outcomes. With the use of biological fluids as a source of lncRNA biomarkers, sample collection ranges from non-invasive (urine, saliva) to minimally invasive (plasma, serum) and invasive (organ biopsy). Due to the properties of lncRNAs, there is pressing need for the standardization of sample collection and the stringent preparation of biomaterials for analysis across different settings, to ensure the specificity and reproducibility of the data [107]. More research on the correlation between various lncRNAs in the same condition and cross-correlation in different diseases is warranted, using extensive data collection and analysis with advanced computational methods and artificial intelligence (AI) approaches.

## 8. Emerging lncRNA Biomarkers of GBM

**ADAMTS9-AS2** (*ADAM metallopeptidase with thrombospondin type 1 motif 9 antisense RNA 2*) is considered a proto-oncogenic GBM lncRNA in most studies [49,241]. This lncRNA is also abnormally upregulated in other malignancies, with significantly increased levels in ovarian cancer tissue [217], while in lung adenocarcinoma, its level is reduced [242]. ADAMTS9-AS2 is involved in several major signalling pathways, including PI3K/AKT and MEK/Erk, and interacts with many miRNAs (in most cases as a sponge) [243]. ADAMTS9-AS2 has both tumour suppressor and proto-oncogenic functions depending on the type of cancer and can be used as a biomarker for cancer. Abnormal expression levels of this lncRNA measured in plasma or tissue have diagnostic value, with changes reported in patients with malignancy of lung, oesophageal and prostate cancer [243], and lung adenocarcinoma [112,244,245]. The decreased expression of ADAMTS9-AS2 in tumour tissue correlates with poor prognosis and shorter survival in patients with oesophageal cancer [246], lung adenocarcinoma [242], breast cancer, and bladder urothelial carcinoma [243]. The expression of ADAMTS9-AS2 was reduced and negatively correlated with the extent of tissue and organ damage, which makes this lncRNA a potential qualitative biomarker [111]. ADAMTS9-AS2 has repeatedly demonstrated GBM oncogenic effects [49,241]. Its expression was measured in resected GBM tissue and cell lines, with levels correlating with glioma grade [241]. Increased ADAMTS-AS2 levels are also prognostic, as higher expression levels were found in GBM patients resistant to TMZ treatment compared to those responding to the same treatment [49]. Considering that elevated levels of this lncRNA are also found in the blood of patients with several diseases, the most effective use of this lncRNA profile in patients with GBM would be in combination with other indicators of disease.

**ANRIL** (*antisense noncoding RNA in INK4 locus*) is considered an oncogenic lncRNA linked to GBM. The dysregulation of ANRIL in blood has been associated with cancers in general, cardiovascular diseases [247], and type 2 diabetes mellitus [248] (see Table 2). This lncRNA can modulate gene expression at the post-transcriptional level by interacting with miRNAs and proteins [249]. Furthermore, ANRIL negatively and positively influences gene expression at the chromatin level [247]. ANRIL functions as a scaffold for PRC2 and, therefore, participates in epigenetic gene silencing [250] and is involved in alternative splicing in HEK293 and HUVEC cells [251]. Through these mechanisms, ANRIL contributes to tumourigenesis processes, increasing cell proliferation, migration, invasion, and metastasis and suppressing apoptosis and senescence [248]. Upregulated ANRIL expression levels have been found to be linked with cancers such as lung, stomach, breast, ovarian, cervical, colorectal, bladder, thyroid, brain, osteosarcoma, myeloma, prostate, endometrial, renal, leukaemia, melanoma, retinoblastoma, and hepatocellular carcinoma [248]. In addition to an increased risk of cancer, polymorphisms in the ANRIL gene are also associated with the risk of atherosclerosis, obesity, and type 2 diabetes. ANRIL expression is also affected by inflammation, with pro-inflammatory factor IFN-γ activating the transcription factor STAT1, thereby inducing ANRIL expression in endothelial cells [249]. Elevated ANRIL can affect the expression of NF-κB-dependent inflammatory molecules, such as IL-6 and IL-8 [250]. In GBM, the oncogenic lncRNA ANRIL is upregulated in cell lines, resected GBM tissue, and the serum of patients diagnosed with glioma [16,54]. The high expression of this lncRNA in patient serum correlates with adverse prognosis, grade, size, and metastasis [54]. This lncRNA should be part of the standard screening procedure of patients with suspected GBM.

**CASC2** (*cancer susceptibility candidate*) is another lncRNA candidate for the diagnosis of GBM [55], with its downregulation also reported in endometrial, lung, gastric, colorectal, and bladder cancer. In clinical practise, low levels of CASC2 are associated with a more aggressive cancer phenotype and shorter survival time [252]. CASC2 is involved in the MAPK and Wnt/B-catenin signalling pathways. This lncRNA functions as a sponge for some oncogenic miRNAs, such as miR-21 and miR-18a [252]. The lncRNA CASC2 was monitored in the blood of patients with type 2 diabetes. Low serum levels of CASC2 predict the appearance of chronic renal failure [139] and rheumatoid arthritis [144] in these patients. Different plasma levels of CASC2, along with IL-6 and IL-8, were found in patients treated for aphthous stomatitis compared to healthy controls. Higher levels of CASC2 after treatment predicted a higher rate of recurrence [135]. CASC2 expression levels measured in whole blood negatively correlate with liver cancer stage [137]. The deregulation of CASC2 expression was also investigated in the serum of patients hospitalized with sepsis. Levels were negatively correlated with the Assessment of Acute Physiology and Chronic Health II (APACHE II) and the Sequential Organ Failure Assessment (SOFA). With lower CASC2 levels, the risk of death increases in these patients. CASC2 insufficiency may be a good biomarker, as it correlates with reduced cytokine release, the severity of multiorgan injury, and prognosis in these patients [144]. On the other hand, the upregulation of CASC2 was observed in pancreatic tissues of patients with acute pancreatitis [253]. The expression of CASC2 was examined in GBM cell lines, xenografts, and tissues resected from patients diagnosed with glioma [55,254,255]. The level of this lncRNA is upregulated compared to healthy controls. This fact leads to changes in the expression of some miRNAs, e.g., miR-193a-5p, and a decrease mTOR expression [254]. The expressional changes negatively correlate with the tumour grade and survival time in patients [255] and with its role in the efficacy of chemotherapy also reported [256]. Unfortunately, data reporting changes in CASC2 expression in the blood of GBM patients are not available. Given the diagnostic and prognostic value of this lncRNA in GBM patients and the significance of changes in blood levels of CASC2 in other diseases, it is desirable to investigate in more detail the dynamics of this tumour suppressor lncRNA in the blood of GBM patients.

**CRNDE** (*colorectal neoplasia differentially expressed*) is an oncogenic lncRNA detected in tissue from GBM patients and is associated with resistance to TMZ therapy [58]. This lncRNA is also abnormally expressed in other cancers. Alterations in CRNDE expression correlate with tumour clinico-pathological characteristics and the prognosis of patients diagnosed with colorectal cancer, breast cancer, cervical cancer, lung adenocarcinoma, multiple myeloma, chronic lymphocytic leukaemia, and ovarian cancer [257]. The physiological expression of CRNDE is tissue-specific; low levels are detected, e.g., in the colorectal mucosa; on the other hand, CRNDE is found in breast tissue and testes in higher amounts [257]. CRNDE interacts with a wide variety of targets involved in the activation of the Wnt/β-catenin signalling pathway, as well as some miRNAs (e.g., miR335-3p) and proteins [258]. CRNDE may serve as a scaffold for some tumour-associated proteins (e.g., DMBT1) [259,260]. Through the molecular mechanisms described above, CRNDE regulates the tumour microenvironment, contributing to tumorigenesis—proliferation, cell invasion, apoptosis, metastasis, and treatment resistance [151]. Elevated levels of this lncRNA are an indicator of the prognosis of cancer patients [151], e.g., in a patient with osteosarcoma [258]. The deregulated expression levels found in the blood of patients hospitalized with sepsis are correlated with APACHE II and SOFA, as well as inflammation, and are a prognostic biomarker for sepsis [151]. Finally, CRNDE appears to be a good biomarker for the clinical course of hepatocellular carcinoma. The available analyses suggest that serum-measured exosomal lncRNA CRNDE is an independent marker of survival time in patients with hepatocellular carcinoma [148]. CRNDE is one of the best characterized lncRNA in association with gliomas and GBM, with increased expression observed in GBM cell lines (including CSCs [261]) and in resected GBM tissues [262]. Tissue expression levels of this lncRNA correlate with prognosis, tumour size and the risk of recurrence [261], and GBM subtype [263], and levels predict patients’ chemosensitivity to TMZ treatment [58]. In vitro experiments suggest that CRNDE knockdown enhances TMZ chemosensitivity in GBM cells [58]. This makes CRNDE a potential therapeutic target for further GBM treatment research. The level of CRNDE in the blood of GBM patients has been investigated [166]. Because it was detected in only 20% of patients, its biomarker potential has not been further investigated [166]. In consideration of the facts described above, it would be a good idea to focus on a larger sample of patients with different subtypes of GBM to see if increased CRNDE expression in the blood of patients indicates only that subtype of GBM or decreased chemosensitivity to TMZ.

**DGCR5** (*DiGeorge syndrome critical region gene 5*) is one of the GBM suppressor lncRNAs [60]. The oncogenic and suppressor functions of DGCR5 have been described depending on the type of malignancy (e.g., gallbladder cancer, lung cancer) [264]. The dysregulation of DGCR5 expression has also been documented in patients with Huntington’s disease [265]. At the molecular level, this lncRNA is involved in various mechanisms of tumourigenesis, including cell proliferation, invasion, migration, apoptosis, and response to therapy. It interacts with many miRNAs, including miR-21, and functions as competing endogenous RNA (ceRNA) [266]. Reduced expression compared to healthy controls has been observed in the following malignancies: cervical [267], laryngeal, bladder [268], pancreatic, thyroid, prostate, ovarian cancer, hepatocellular carcinoma [269,270], colorectal cancer [266], and gliomas [60]. For these types of malignancies, DGCR5 could be used as a biomarker, as reduced expression levels correspond to clinical stage, tumour size, survival time, and the amount of metastasis [264,266,271,272]. On the other hand, increased expression was detected in gallbladder cancer and triple negative breast cancer [264,273]. This lncRNA also correlates with the number of immune cells and the strength of the immune response in the tumour microenvironment [264]. DGCR5 is downregulated in glioma tissue and cell lines [60]. The analysis of data from xenograft experiments confirmed that this lncRNA acts as a tumour suppressor by inhibiting glioma growth [60]. The level of lncRNA in resected tissue negatively correlates with glioma grade and prognosis [274]. This lncRNA could be included in the tissue signature of GBM to refine diagnosis and prognosis. DGCR5 expression level correlates with the amount of immune and stromal cells and is, thus, associated with immune response and immune infiltration [274]. Further studies indicate that this lncRNA is involved in the process of angiogenesis and could be a promising therapeutic target [275]. Given its diagnostic and prognostic character in GBM and deregulation in the blood of gastric cancer patients [155], this lncRNA is an interesting target for analysis in the blood of GBM patients.

**GAS5** (*growth arrest specific 5*) is a GBM tumour suppressor lncRNA [63]. The decreased expression of this lncRNA is also detected in other cancers including breast, prostate, ovarian, cervical, colorectal, gastric, kidney, bladder, lung, pancreatic, endometrial, and renal cancers, as well as melanoma, osteosarcoma, neuroblastoma, and gliomas [276]. GAS5 naturally accumulates in cells after growth arrest induced by, for example, nutrient deficiency. GAS5 affects cell cycle progression, and it is necessary for normal cell growth arrest. High levels of GAS5 expression inhibit cell cycle progression, while decreased GAS5 expression reduces apoptosis and promotes accelerated cell division [276]. GAS5 is considered a tumour-suppressive lncRNA in association with many malignancies, in which the reduced expression of this transcript has been detected. Clinico-pathological characteristics, which include survival time, relapse-free survival, the presence of distant metastases, the presence of lymph node metastases, tumour size, and progression, are inversely correlated with expression levels in different types of cancer, suggesting that GAS5 could become a diagnostic and prognostic biomarker. Furthermore, it also has the potential to be a biomarker allowing for the monitoring of therapeutic responses [277]. GAS5 tumour suppression has been associated with gliomas, and the expression level of this lncRNA is correlated with the degree of tumour malignancy and patient survival time. Differential expression levels of GAS5 are detected not only in tissues but also in body fluids, including blood and urine [278]. Decreased plasma and serum GAS5 levels have been detected in patients with multiple sclerosis and in patients with myelofibrosis. Measured values were correlated with the clinico-pathological status of the patient [161,162]. Reduced expression was detected in serum from patients with various diseases including breast cancer [167], stroke [168], COVID-19 [170], liver cancer, sepsis [181], rheumatoid arthritis, and osteoporosis [177]. The tumour suppression of GAS5 has been associated with gliomas, and the expression level of this lncRNA correlates with the degree of tumour malignancy and patient survival time. GAS5 transcription is higher in lower-grade gliomas compared to higher-grade gliomas, including GBM [279]. Low levels of GAS5 expression observed in GBM compared to healthy controls correlate with poor prognosis [269]. Serum levels of GAS5 may become a good prognostic biomarker as part of the lncRNA signature because deregulated levels of this lncRNA are associated with the two-year overall survival of GBM patients after surgery [166]. The deregulation of GAS5 in multiple diseases shows the importance of this lncRNA. For the clinical use of GAS5 as a biomarker, specific sets of lncRNAs are needed to facilitate higher diagnostic specificity.

**LINC00467** (*long intergenic non-protein coding RNA 467*) is an oncogenic GBM lncRNA, and its expression correlates with the grade of glioma [72]. This lncRNA has been shown to be pro-inflammatory in association with some other malignancies such as gastric cancer, with its increase reported in lung cancer, breast cancer, colorectal cancer, hepatocellular carcinoma, osteosarcoma, head squamous cell carcinoma, and others [280]. LINC00467 is part of several signalling pathways including Akt, STAT, and EGFR, and its deregulation may contribute to pro-inflammatory mechanisms [281,282]. Tumourigenesis can also occur through the sponging of, e.g., miR-4779 and miR-7978 [283]. LINC00467 can also act as a ceRNA and, thus, participate in the regulation of signalling pathways (e.g., EGFR) and tumorigenesis [284]. It correlates with the clinical stage of various cancer types, with their poor prognosis and survival time [281,285,286,287,288]. Interestingly, this lncRNA can encode a short ASAP peptide. Research shows that this micropeptide is involved in mitochondrial metabolism, and high levels correlate with a poor prognosis in patients with colorectal cancer [289]. Another argument for considering this lncRNA among diagnostic and prognostic biomarkers, including GBM, is the detection of LINC00467 deregulation in the plasma of patients with acute myeloid leukaemia [101]. Increased expression levels of LINC00467 have also been detected in prostate cancer tissue. The level of expression varied between cells and specifically between two macrophage phenotypes, pro-inflammatory and anti-inflammatory. Studies show that LINC00467 is involved in the polarization of macrophages towards the pro-inflammatory type. These facts make LINC00467 a promising therapeutic target for patients with early stage prostate cancer [290]. LINC00467 was analysed in glioma tissues and in cell lines [291]. The expression level was upregulated [291], and its knockdown inhibited the proliferation of cell [291] and induced apoptosis [291]. These observations make LINC00467 a potential therapeutic target. More experiments, including the analysis of this lncRNA in patients’ blood, are required to designate LINC00467 as a GBM biomarker.

**LINC00641** (*long intergenic non-protein coding RNA 641*) is a potential biomarker for GBM and is differentially expressed in other types of cancer [73,292]. LINC00641 can be classified as both a tumour suppressor and an oncogenic lncRNA depending on the type of cancer. The tissue expression of this biomolecule is upregulated in association with gastric, renal, prostate, and rectal cancers and acute myeloid leukaemia [292,293]. On the other hand, reduced expression levels are linked with cervical, bladder, breast, non-small cell lung, and thyroid cancer [292]. Differences in tissue expression are associated with prognosis and survival in patients with cancers that include prostate cancer, thyroid cancer, bladder cancer [294], gastric cancer, renal cell carcinoma, and rectal cancer [292,295]. In patients with breast cancer, expression levels correlate with tumour size and clinical stage, including lymph node metastasis [295]. LINC00641 interacts with many miRNAs as a sponge, e.g., for miR-197-3p, or as competing endogenous RNAs (ceRNAs) in cervical, bladder, and rectal cancers and acute myeloid leukaemia [292]. LINC00641 is also involved in the regulation of several signalling pathways including PTEN/PI3K/AKT and Notch-1 [296]. Therefore, targeting the LINC00641/miR-197-3p/KLF10/PTEN/PI3K/AKT cascade could hold promise as a therapeutic strategy. LINC00641 has been shown to be involved in the regulation of proliferation and apoptosis, as well as invasion and metastasis in several cancer types. Many studies in cell lines demonstrated the therapeutic potential of LINC00641. Targeting this lncRNA has been reported to alleviate features of tumourigenesis in almost all cancer types mentioned above [292]. In some cases, for example, in gastric cancer, targeting LINC00641 also affects drug resistance [297]. The above facts nominate LINC00641 as a high-quality potential biomarker and therapeutic target in the context of cancer and other diseases. LINC00641 has also been detected at higher levels in the serum of patients diagnosed with inflammatory bowel diseases such as ulcerative colitis and Crohn’s disease and may, thus, be part of a non-invasive diagnostic pathway [204]. Furthermore, the results of some studies suggest the involvement of this lncRNA in the autophagy process and its indirect effect on the expression of brain-derived neurotrophic factor (BDNF) [298,299]. It is reportedly downregulated in GBM cell lines and tissues and is proposed to be a tumour suppressor lncRNA acting as an inhibitor of GBM cell proliferation [300,301]. Based on bioinformatics analyses, LINC00641 is recommended as part of the lncRNA signature for more accurate diagnosis from resected tissue [300,302]. In the serum of GBM patients, LINC00641 was analysed together with LINC00565, and both lncRNAs showed increased expression compared to healthy controls [73].

**MIR210HG** (*MIRNA210 host gene*) is an oncogenic GBM lncRNA [45] and is deregulated in other cancers (liver, lung, pancreatic, breast, gastric, cervical, ovarian, and colorectal) and non-cancerous diseases such as preeclampsia, acute renal injury, and others [303]. MIR210HG is involved in cell proliferation, migration and invasion, energy metabolism, autophagy, hypoxia, radiosensitivity, and chemoresistance. MIR210HG has been confirmed to interact with many miRNAs, e.g., by sponging miR-520a-3p, suppressing trophoblast migration and invasion in vitro, or as a ceRNA [304]. Cell culture studies and clinical data show that there is an association between drug resistance and the abnormal expression of MIR210HG in certain cancers, including GBM, pancreatic cancer, non-small cell lung cancer, ERPR/Her2-type breast carcinoma, and colorectal cancer. These data make MIR210HG a novel therapeutic target that could improve tumour sensitivity to radiotherapy and chemotherapy and inhibit neoplastic process [45,303,305]. Elevated levels of MIR210HG expression have been found in other cancers, with its higher levels reported in hepatocellular carcinoma tissue, which correlates with the clinical stage of the disease and tumour characteristics, including size, vascular invasion, and histological differentiation. It is also negatively correlated with overall cancer survival and could, therefore, be a good prognostic marker [306]. The abnormal expression of MIR210HG was found in pancreatic tumour tissue and its level is also associated with the survival time of patients [307]. Expressional changes are also associated with the clinical presentation of patients with osteosarcoma, as well as breast cancer, colorectal cancer, gastric cancer, cervical cancer, and ovarian cancer [303]. Interestingly, this lncRNA is overexpressed in the sperm of infertile men with varicocele and negatively correlates not only with the quantity of sperm but also with the motility of the sperm [308]. MIR210HG was detected in the placenta of patients with preeclampsia compared to healthy controls [304]. These data suggest that MIR210HG may be included in the list of potential prognostic markers related to various diseases. The upregulation of MIR210HG was detected in the serum of glioma patients compared to healthy controls [227]. In addition to blood, MIR210HG levels were also elevated in GBM tissue and cell lines [45]. As mentioned above, MIR210HG is involved in the mechanism of hypoxia, which affects tumour aggressiveness. Elevated levels of this lncRNA predict poor prognosis associated with cell invasion, CSC, and TMZ resistance [45]. Based on bioinformatics analyses, MIR210HG was found to be part of a set of lncRNAs that can be used to distinguish GBM from other gliomas [309].

**ZEB1-AS1** (*zinc finger E-box-binding homeobox 1 antisense 1*) is an oncogenic GBM lncRNA [79]. Protumour ZEB1-AS1 is associated with several other malignancies, including colorectal cancer, breast cancer, gastric cancer, prostate cancer, hepatocellular carcinoma, non-small cell lung cancer, osteosarcoma, and others [310]. ZEB1-AS1 may potentially boost the proliferation, invasion, and migration capabilities of melanoma cells by directly suppressing miR-1224-5p. A study showed that elevated levels of ZEB1-AS1 were correlated with a decrease in the overall survival rate among melanoma patients, suggesting that ZEB1-AS1 and miR-1224-5p play crucial roles in melanoma pathogenesis and could serve as predictive biomarkers and potential therapeutic targets [311]. ZEB1-AS1 lncRNA also plays a role in non-malignant conditions such as atherosclerosis, pulmonary fibrosis, ischemic vascular disease, and complications accompanying diabetes, including diabetic nephropathy [103]. ZEB1-AS1 is involved in the regulation of gene expression and, thus, contributes to cancer cell proliferation and migration [310]. It is an important modulator of ZEB1 gene expression, which is one of the main regulators of the epithelial–mesenchymal transition. ZEB1-AS1 acts as a sponge for many miRNAs and can, therefore, influence other genes [103]. Based on the results of the studies, ZEB1-AS1 appears to be a good biomarker not only in the context of cancer. Measured serum and plasma levels correlate with prognosis, response to treatment and stage in the following diseases [103]. In the context of colorectal cancer, ZEB1-AS1 has a diagnostic function. Its expression level correlates with clinical stage and histological grade, metastasis, and microvascular invasion, and its overexpression is associated with a poor prognosis [310]. Serum expression levels of ZEB1-AS1 were measured in patients with oral squamous cell carcinoma before and after tumour resection, with highly detectable differences. Data from this study suggest that ZEB1-AS1 could be a good marker for measuring treatment success [311]. In patients treated for oesophageal cancer, ZEB1-AS1 was also detected in serum. The measured levels were correlated with a poor prognosis and ZEB1-AS1 levels in tumour tissues. Clinical studies show that ZEB1-AS1 expression levels also correlate with complications of diabetes [312]. Different expression levels of this lncRNA were measured in plasma from patients treated for diabetes-related complications (e.g., lung damage, nephropathy) compared to diabetics without complications and a healthy group, where the expression was higher [240]. The opposite trend was observed in the serum of patients with atherosclerosis, where the increased expression of ZEB1-AS1 was detected compared to healthy controls [313]. Changes in ZEB1-AS1 expression were detected in glioma tissue (including GBM) and GBM cell lines [314]. In both types of material, ZEB1-AS1 is detected at high levels, and in resected tissue, it correlates with tumour size and malignancy grade (I–IV) [314]. In vitro experiments suggest that the knockdown of this lncRNA induces G0/G1 phase arrest and correspondingly reduces the percentage of cells in S phase, thus affecting GBM cell proliferation, invasion, and migration [314]. Given the diagnostic relevance of ZEB1-AS1 to GBM and its tissue-related changes, as well as deregulation in the blood of patients with other diseases, the analysis of the ZEB1-AS1 profile and dynamics in the blood of GBM patients is well-warranted.

## 9. Conclusions

LncRNAs, which are the largest group of noncoding transcripts, have received much research and translational interest. Their specificity for various tissue types and changes under different physiological and pathophysiological conditions have been explored as markers for normal and disease states. Most human diseases, including different forms of cancer, are linked to deregulated lncRNAs, making these molecules promising biomarker candidates and therapeutic targets. Advances in genome and transcriptome analysis have facilitated lncRNA research with numerous new transcripts identified and characterized over recent years. Several databases specialized in lncRNAs have been created to organize and use the growing information, some of which were utilized in the writing of this article (e.g., LNCipedia 5.2, lncRNAfunc).

As emerges from this analysis of rapidly advancing research on lncRNAs of the brain specimens and biological fluids, these transcripts can be highly instructive for the diagnosis of GMB, which belongs to the most aggressive group of malignant brain tumours in adults and resists conventional therapies. Despite the relatively low incidence (3–4 cases per 100,000 people), GMB remains one of the greatest challenges and priorities for neuro-oncology and cancer research in general, owing to its severity and high mortality. The effectiveness of treatment and the course of disease is influenced by the heterogeneity of tumour tissue, with their regulation involving lncRNAs, as shown in many studies. Importantly, changes in lncRNA profiles in the blood of GBM patients provide reliable readouts of the state and grade of pathology, offering a rapid and lowly invasive diagnostic approach. Based on the analysis of clinical and translational data, we propose diagnostic lncRNA fingerprint for GBM, which combines ANRIL (↑), HOTAIR (↑), LINC00641 (↑), LINC00565 (↑), MALAT1 (↑), SAMMSON (↑), and GAS5 (↓). Given that lncRNAs profiling in blood involves relatively simple sample collection and measurement procedures, their in-depth profiling could lead to early interventions with better therapeutic outcomes.

## Figures and Tables

**Figure 1 biomedicines-12-00932-f001:**
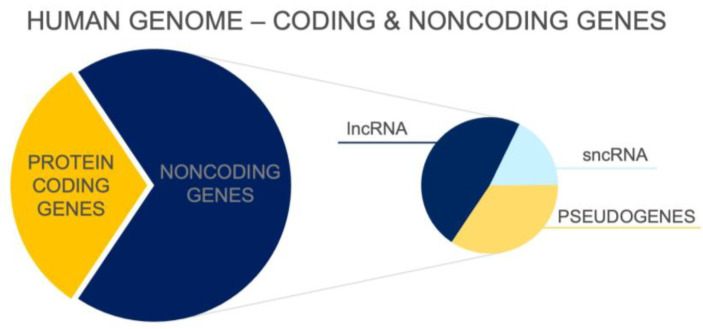
Distribution of coding and noncoding genes in the human genome according to ENCODE Release version 45 [6]. lncRNA—long noncoding RNA; sncRNA—small noncoding RNA.

**Figure 2 biomedicines-12-00932-f002:**
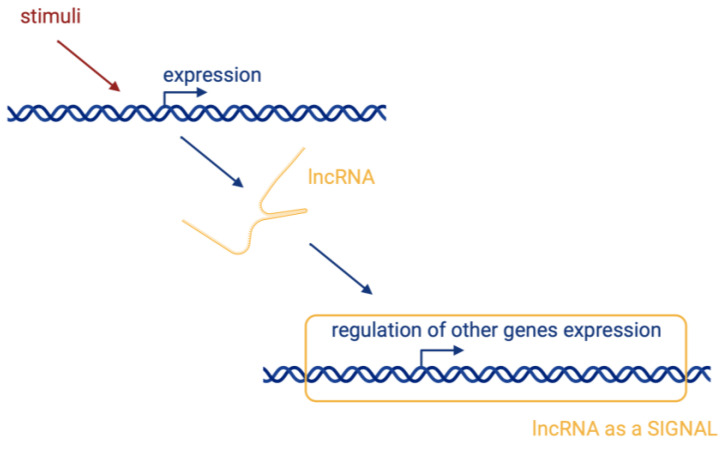
Signal—lncRNAs are transcribed at a specific site at a specific time. Their transcription is cell type-specific, inducing an active signalling event. Created with BioRender.com (accessed on 21 March 2024).

**Figure 3 biomedicines-12-00932-f003:**
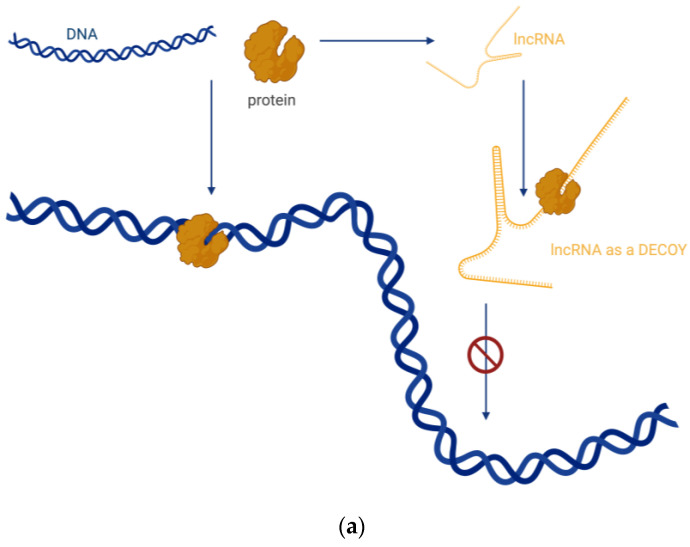
(**a**) Decoy—transcripts of lncRNAs serve as decoys for target proteins. (**b**) Sponge—specific lncRNAs act as sponges for some miRNAs. Created with BioRender.com (accessed on 21 March 2024).

**Figure 4 biomedicines-12-00932-f004:**
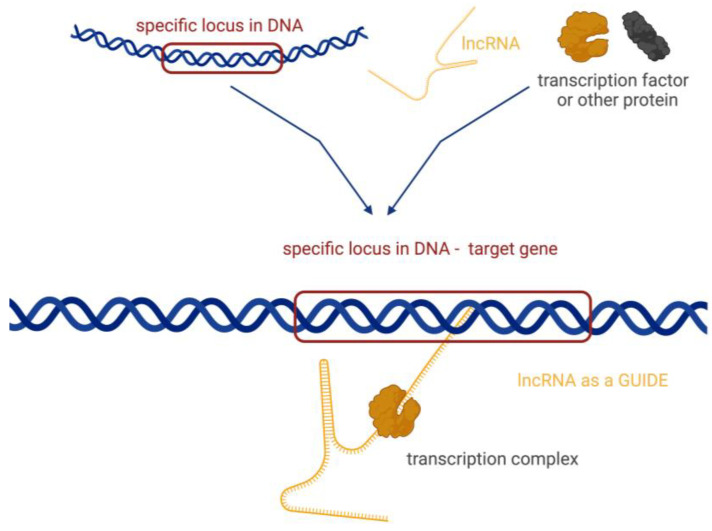
Guides—lncRNA molecules control the placement of ribonucleoprotein complexes at specific target sites. Created with BioRender.com (accessed on 21 March 2024).

**Figure 5 biomedicines-12-00932-f005:**
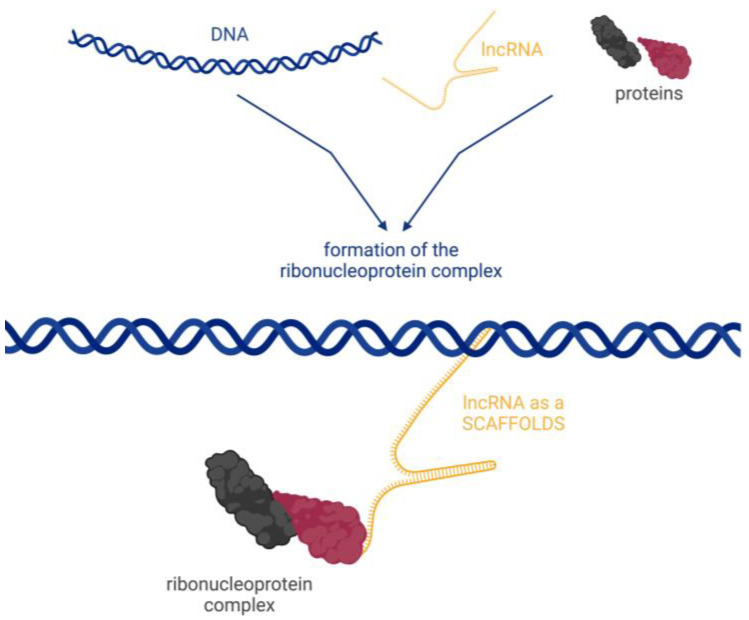
Scaffolds—lncRNA is a scaffold for other molecules that can bind to a given lncRNA to form a ribonucleoprotein complex. Created with BioRender.com (accessed on 21 March 2024).

**Figure 6 biomedicines-12-00932-f006:**
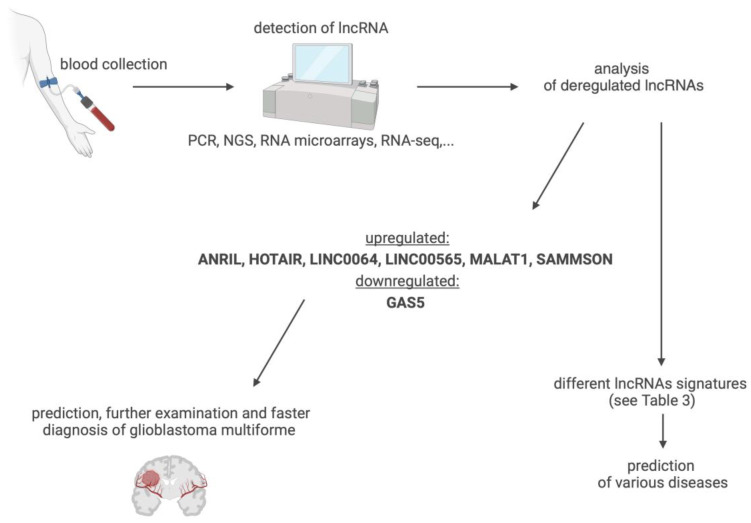
Suggested lncRNA signatures for glioblastoma multiforme and clinical application. Created with BioRender.com (accessed on 21 March 2024).

**Table 1 biomedicines-12-00932-t001:** Systematic list of long noncoding RNAs (lncRNAs) associated with glioblastoma multiforme (GBM). This list includes lncRNAs whose expression is deregulated in association with GBM in vitro, in vivo, and ex vivo—in GBM cell lines, in GBM primary tissue, in xenografts, in GBM tissue from resected tumours, and in blood from GBM patients. In addition to their common names, lncRNAs are also identified by an Ensembl tag and by specifying their position on chromosomes. Abbreviations: lncRNA—long noncoding RNA, GBM—glioblastoma multiforme, TMZ—temozolomide, CSC—cancer stem cells, GSC—glioblastoma stem cells, TCGA—The Cancer Genome Atlas Program.

lncRNA Name /Alternative Transcript Name/	Gene Location Class Ensembl Gene ID	Role in GBM	Expression	Function in GBM	Data	Ref.
AC016405.3 /RP11-44N11.2/ /lnc-DERL1-3/	8q24.13 bidirectional ENSG00000272384	suppressor	↓	suppressing proliferation and invasion	clinical association; GBM primary tissue; GBM cell line	[33]
ADAMTs9-AS2 /NONHSAT090261/	03p14.1 antisense ENSG00000241684	oncogenic	↑	TMZ resistance	clinical association; GBM cell line	[49]
AGAP2-AS1 /HSALNG0091650/	02q14.1 antisense ENSG00000255737	oncogenic	↑	proliferation, viability	GBM primary tissue; GBM cell line	[52]
AHIF /lnc-TMEM30B-9/ /HIFiA-AS2/	14q23.2 antisense ENSG00000258777	protumour	↑	invasion, viability, GSC, radiation resistance	GBM cell line; GSC mesenchymal line	[53]
ANRIL /CDKN2B-AS1/	09p21.3 antisense ENSG00000240498	oncogenic	↑	cell proliferation	GBM cell line; GBM tissue; GBM patient serum	[13,54]
lncRNA-ATB	14q11.2 intronic -	protumour	↑	invasion of cell	GBM cell line	[40]
CASC2	10q26.11 antisense ENSG00000177640	suppressor	↓	inhibitor of proliferation	GBM cell line; GBM tissue; xenograft	[55]
CASC7 /lnc-AGO2-1/	8q24.3 intronic ENSG00000259758	suppressor	↓	inhibitor proliferation, regulation of cell cycle	GBM primary tissue; GBM cell line	[43]
CASC9 /LINC00981/ /RP11-697M17.1-003/	8q21.13 intronic ENSG00000249395	oncogenic	↑	tumourigenesis	GBM cell line	[56]
CCND2-AS1	12p13.32 antisense ENSG00000255920	protumour	↑	proliferation and growth	GBM cell line; GBM patient tissue	[57]
CRNDE /lnc-IRX3-80/	16q12.2 antisense ENSG00000245694	oncogenic	↑	proliferation, invasion, migration, inhibition of apoptosis	GBM cell line; GBM patient tissue	[58]
DCST1-AS1	01q21.3 antisense ENSG00000232093	protumour	↑	proliferation	clinical related; GBM primary tissue; primary cultivation	[59]
DGCR5	22q11.21 antisense ENSG00000237517	suppressor	↓	proliferation, migration, invasion, apoptosis	GBM cell line; GBM tissue	[60]
DLEU1-AS1	13q14.3 intronic ENSG00000186047	biomarker	↑	proliferation, cell cycle, autophagy; correlation with prognosis	GBM cell line; GBM tissue	[61]
ECONEXIN /LINC00461/	05q14.3 intronic ENSG00000245526	protumour	↑	proliferation	GBM cell line; GBM tissue; TCGA data	[62]
FAM66C	12p13.31 antisense ENSG00000226711	-	↑↓	tumour microenvironment	GBM cell line; GBM tissue; TCGA data	[44]
GAS5	01q25.1 antisense ENSG00000234741	suppressor	↓	inhibition of proliferation, invasion and viability	GBM cell line; GBM tissue; GBM patient serum	[63]
H19 /D11S813E/ /ASM1/	11p15.5 intronic ENSG00000130600	protumour	↑	proliferation, invasion, angiogenesis	GBM cell line	[64]
HMMR-AS1	05q34 antisense ENSG00000251018	protumour	↑	tumourigenesis, proliferation, invasion, radiation resistance	GBM cell line	[50]
HOTAIR	12q13.13 antisense ENSG00000228630	protumour	↑	proliferation, invasion, therapy resistance, chromatin remodelling	clinical association; GBM patient tissue/serum; cell line; xenoimplants	[65,66]
HOTAIRM1 /HOXA-AS1/	07p15.2 antisense ENSG00000233429	oncogenic	↑	proliferation, invasion, viability	clinical association TCGA; GBM primary tissue; GBM cell line	[67]
HOXA-AS2	07p15.2 antisense ENSG00000253552	protumour	↑	migration, invasion, viability	GBM tissue; GBM cell line	[68]
HOXB13-AS1 /lncHOXB13-1/	17q21.2 intronic ENSG00000159184	protumour	↑	proliferation, progress	GBM tissue; GBM cell line	[69]
HOTTIP /HOXA-AS6/	07p15.2 antisense ENSG00000243766	antitumour	↓	inhibition of cell cycle, induction of apoptosis	GBM tissue; GBM cell line	[70]
HULC /lnc-BMP6-106/	06p24.3 intronic ENSG00000285219	protumour	↑	proliferation, angiogenesis, activity of MGMT	GBM cell line; GBM patient tissue	[29]
KTN1-AS1	14q22.3 antisense ENSG00000186615	tumour suppressor	↑	viability and invasion of cell; correlation with prognosis	GBM tissue; GBM cell line; TCGA data	[71]
LINC00467 /NR_026761/	01q32.3 intronic ENSG00000153363	protumour	↑	proliferation and invasion	GBM cell line	[72]
LINC00565	13q34 intronic ENSG00000260910	unknown; biomarker	↑	correlation with prognosis	GBM patient serum	[73]
LINC00641	14q11.2 intronic ENSG00000258441	unknown; biomarker	↑	correlation with prognosis	GBM patient serum	[73]
LINC01393	07q31.2 intronic ENSG00000225535	unknown; biomarker	↑	tumour progress; correlation with prognosis	GBM tissue; GBM cell line; TCGA data	[74]
LINC01426	21q22.12 intronic ENSG00000234380	oncogenic	↑	proliferation, invasion, viability	clinical association TCGA; GBM primary tissue; GBM cell line	[75]
LINC01446	07p12.1 intronic ENSG00000205628	protumour	↑	tumourigenesis, progress	clinical association; GBM cell line; xenografts	[76]
LINC01494	02q35 intronic ENSG00000228135	oncogenic	↑	proliferation, invasion	clinical association; GBM tissue; GBM cell line	[77]
LINC01503	9q34.11 intronic ENSG00000233901	oncogenic; biomarker	↑	migration, invasion, apoptosis; correlation with malignancy grade and prognosis	GBM tissue; GBM cell line; TCGA data	[78]
LINC01711	20q13.32 intronic ENSG00000268941	protumour	↑	proliferation, migration, invasion correlation with prognosis	GBM tissue; GBM cell line	[79]
LINC02283	04q12 intronic ENSG00000248184	oncogenic	↑	correlation with expression of PDGFRA, malignancy	patient GSC lines; xenoimplants; GBM tissue	[80]
LINC-ROR /ROR/	18q21.31 intronic ENSG00000258609	unknown	↑↓	GSC	GBM tissue; GBM cell line	[81]
lnc-TALC /LNCARSR/ /linc-GNAQ-7/	09q21.31 intronic ENSG00000233086	protumour	↑	TMZ resistance, tumour relapse	TMZ-selected GBM cell lines	[48]
MAFG-DT /MAFG-AS1-001/	17q25.3 intronic ENSG00000265688	protumour	↑	proliferation	GBM tissue; GBM cell lines	[82]
MALAT1	11q13.1 intronic ENSG00000251562	unknown	↑↓	invasion, proliferation, migration, apoptosis, permeability of BBB, chemosensitivity	clinical association; GBM patient tissue and serum; GBM cell lines; xenografts	[38]
MATN1-AS1	01p35.2 intronic ENSG00000186056	suppressor	↓	inhibition of proliferation and invasion	GBM primary tissue lines; GBM cell lines	[38]
MDC1-AS	06p21.33 antisense ENSG00000224328	suppressor	↓	inhibition of proliferation	GBM cell lines	[83]
MEG3 /lnc-DLK1-3/	14q32.2 intronic ENSG00000214548	suppressor	↓	inhibition of proliferation	GBM tissue; GBM cell lines	[84]
MIAT	22q12.1 intronic ENSG00000225783	oncogenic	↑	proliferation, migration, metastasis	GBM tissue; GBM cell lines	[37]
MIR210HG	11p15.5 intronic ENSG00000247095	unknown; biomarker	↑	hypoxia, invasion, TMZ resistance, correlation with prognosis	GBM cell line; xenografts; TCGA data; GBM patient plasma	[45]
MNX1-AS1 /CCAT5/ /LOC645249/	07q36.3 intronic ENSG00000243479	oncogenic	↑	proliferation, migration, invasion	GBM tissue; GBM cell lines	[85]
NCK1-AS1 /SLC35G2-AS1/ /NCK1-DT/	03q22.3 antisense ENSG00000239213	protumour	↑	TMZ resistance	primary tissue; GBM cell lines	[86]
NEAT1 /LINC00084/	11q13.1 intronic ENSG00000245532	protumour	↑	proliferation, glycolysis	GBM primary tissue; cell lines; xenografts	[87]
PART1	05q12.1 antisense ENSG00000152931	tumour suppressor	↓	inhibition of progression and tumour growth	clinical association TCGA; GBM tissue; GBM cell lines	[88]
PARTICL /PARTICLE/	2p11.2 circulating ENSG00000286532	regulation of tumour suppressors	-	tumour microenvironment, chromatin dynamics	GBM cell lines; GBM tissue	[32,89]
PCAT1 /PCA1/	08q24.21 intronic ENSG00000253438	unknown	↑↓	viability, DNA repair	GBM cell lines	[42]
PVT1 /lncRNA1331/	08q24.2 intronic ENSG00000249859	oncogenic	↑	tumourigenesis, progress	GBM tissue; GBM cell lines; xenoimplants	[90]
RBPMS-AS1	08p12 antisense ENSG00000254109	antitumour	↓	radiosensitivity, apoptosis	GBM tissue; GBM cell line; xenoimplants	[91]
RPSAP52	12q14.3 antisense ENSG00000241749	unknown; biomarker	↑	correlation with prognosis	clinical association; GBM primary tissue; GBM cell line	[92]
RUNX1-IT1	21q22.12 intronic ENSG00000159216	protumour	↑	cell cycles, proliferation	GBM tissue; GBM cell line	[93]
SAMMSON /LINC01212/	03p13 intronic ENSG00000240405	oncogenic; potential biomarker	↑	proliferation, viability, invasion, apoptosis	GBM tissue; GBM cell line; GBM patient serum	[94]
SOX2-OT	3q26.3 overlapping ENSG00000242808	unknown; biomarker	↑	migration and invasion; correlation with prognosis	GBM tissue; GBM cell lines	[87]
TALNEC2 /LINC01116/	02q31.1 intronic ENSG00000163364	protumour	↑	tumourigenesis, radiation resistance	clinical association TCGA; GBM primary tissue; GBM cell lines	[95]
TP73-AS1 /lnc-LRRC47-78/ /KIAA0495/	01p36.32 antisense ENSG00000227372	unknown; biomarker	↑	correlation with prognosis; resistance and metabolism, TMZ in GSC	clinical association TCGA; GSC lines	[96]
TSLC1-AS1 /lnc-NXPE2-1/ /RP11-713B9/	11q23.2 antisense ENST00000546273	tumour suppressor	↓	inhibition of cell proliferation, migration and invasion	GBM tissue; GBM cell lines	[97]
TUSC7 /LINC00902/	03q13.31 antisense ENSG00000243197	tumour suppressor	↓	inhibition, resistance to TMZ, tumour malignancy	GBM cell line; GBM patients resistant to TMZ tissue	[98]
TUG1	22q12.2 antisense ENSG00000253352	unknown	↑↓	permeability of BBB	GBM tissue; GBM cell line	[46]
TUNAR	14q32.2 intronic ENSG00000250366	unknown	↑	regulation of tumour progress, cell cycles	GBM cell lines	[99]
UCA1 /UCAT1/ /oncolncRNA-36/	19p13.12 intronic ENSG00000214049	protumour	↑	proliferation, invasion, migration; glycolysis	GBM tissue; GBM cell lines	[100,101]
XIST	Xq13.2 intronic ENSG00000229807	protumour	↑	permeability of BBB, angiogenesis, proliferation of CSC, migration, invasion	GBM tissue; GBM cell line	[102]
ZEB1-AS1	10p11.22 antisense ENSG00000237036	protumour	↑	cell proliferation, migration, invasion	GBM cell line	[103]
ZBED3-AS1	05q13.3 antisense ENSG00000250802	unknown; biomarker	↓	TMZ resistance	TMZ-resistant GBM cell line a tissue	[104]

**Table 2 biomedicines-12-00932-t002:** Systematic list of potential GBM lncRNA biomarkers and their deregulation in blood in other diseases. Putative lncRNA biomarkers of GBM (Table 1) are also deregulated in the blood of patients with other diseases.

lncRNA	Expression	Disease	Expression Level Correlates	Fluids	Ref.
ADAMTs9-AS2	↓	ischemic stroke	severity of disability	plasma	[111]
	↓	non-small cell lung cancer	aggressive tumour behaviour	serum	[112]
ANRIL	↑	breast cancer	metastasis	serum	[113]
	↑	coronary artery disease	prognosis, degree of inflammation, severity of disability	plasma	[114]
	↑	COVID-19	severity of disability	blood	[115]
	↑	Crohn’s disease	diagnosis	serum	[116]
	↑	diabetes mellitus	diagnosis	serum	[117]
	↑	glioma	tumour grade and prognosis	serum	[54]
	↑	intraductal papillary mucinous neoplasms of the pancreas	malignant prediction	plasma	[118]
	↑	ischemic stroke	severity of disability	serum	[119,120]
	↑	multiple myeloma	prognosis	plasma	[121]
	↑	neonatal sepsis	higher risk of mortality	plasma	[122]
	↑	non-small cell lung cancer	prognosis	serum, plasma	[123,124]
	↑	pituitary adenomas	prognosis	plasma	[125]
	↑	sepsis	severity of disability and prognosis	plasma	[126]
	↑	stable angina	level of troponin 1	plasma	[127]
	↑	ulcerative colitis	diagnosis	serum	[116]
	↓	acute exacerbation of chronic obstructive pulmonary disease	levels of inflammatory cytokines	plasma	[128]
	↓	acute ischemic stroke	clinico-pathological symptoms	plasma	[129]
	↓	glaucoma	clinico-pathological symptoms	serum	[130]
	↓	multiple sclerosis	diagnosis	blood	[131]
	↓	paediatric inflammatory bowel disease	diagnosis	serum	[132]
	↓	preeclampsia	diagnosis	serum	[133]
CASC2	↑	osteoarthritis	level of IL-17	plasma	[134]
	↑	aphthous stomatitis	level of IL-6 and IL-18plasma	plasma	[135]
	↓	diabetic nephropathy	diagnosis	serum	[136]
	↓	hepatocellular carcinoma	tumour grade	serum	[137]
	↓	childhood asthma	diagnosis	serum	[138]
	↓	chronic renal failure	diagnosis	serum	[139]
	↓	oral squamous	prognosis	plasma	[140]
	↓	rheumatoid arthritis	diagnosis	serum, plasma	[141,142]
	↓	sepsis	clinico-pathological symptoms	serum, blood	[143,144]
CRNDE	↑	acute myeloid leukaemia	clinico-pathological symptoms	blood	[145]
	↑	colorectal carcinoma	aggressive tumour and liver metastasis	serum, plasma	[146,147]
	↑	hepatocellular carcinoma	tumour size and differentiation	serum	[148]
	↑	nasopharyngeal carcinoma	lymph node metastasis	serum	[149]
	↑	non-small cell lung cancer	diagnosis	plasma	[150]
	↑	sepsis	severity of disability	serum	[151]
	↑	severe pneumonia	prognosis	serum	[152]
	↓	chronic lymphocytic leukaemia	prognosis	serum	[153]
	↓	sepsis	increasing levels after treatment	plasma	[154]
DGCR5	↓	gastric cancer	clinico-pathological symptoms, metastasis	plasma	[155]
	↓	hepatocellular carcinoma	diagnosis	serum	[156]
DLEU-AS1	↑	diabetic foot ulcer	diagnosis	serum	[157]
	↑	endometrial cancer	clinico-pathological symptoms	serum	[158]
GAS5	↑	atherosclerosis	diagnosis	serum	[159]
	↑	malignant mesothelioma	diagnosis	plasma	[160]
	↑	multiple sclerosis	clinico-pathological symptoms	serum	[161]
	↑	myelofibrosis	clinico-pathological symptoms	plasma	[162]
	↑	osteoporosis	diagnosis	plasma	[163]
	↑	osteoporosis with fractures	upregulated in the presence of a fracture	serum	[164]
	↑	polycystic ovary syndrome	diagnosis	plasma	[165]
	↓	glioblastoma multiforme	prognosis	serum	[166]
	↓	breast cancer	diagnosis	serum	[167]
	↓	cerebrovascular stroke	diagnosis	serum	[168]
	↓	coronary artery disease	diagnosis	plasma	[169]
	↓	COVID-19	severity of disability	serum	[170]
	↓	diabetes mellitus 2	diagnosis	serum	[171]
	↓	hepatocellular carcinoma	diagnosis	plasma	[172]
	↓	chronic hepatitis B virus infection	liver fibrosis	serum	[173]
	↓	mycobacterium tuberculosis	diagnosis	serum	[174]
	↓	non-small cell lung cancer	tumour size and metastasis	serum, plasma	[175,176]
	↓	osteoporosis	diagnosis	serum	[177]
	↓	polycystic ovary syndrome	biomarker of insulin resistance	serum	[178]
	↓	rheumatoid arthritis	diagnosis	serum, plasma	[179,180]
	↓	sepsis	diagnosis	serum	[181]
	↓	systemic lupus erythematosus	diagnosis	plasma	[182]
HOTAIR	↑	Alzheimer’s disease	clinico-pathological symptoms, decreasing levels after treatment (exercises)	serum	[183]
	↑	breast cancer	lymph node metastasis	plasma, serum	[184,185]
	↑	colorectal carcinoma	diagnosis	plasma	[186]
	↑	congenital heart diseases	diagnosis	plasma	[187]
	↑	coronary artery disease	diagnosis	blood	[188]
	↑	diabetes mellitus 2	complications of diabetes	serum	[189]
	↑	oesophageal squamous cell carcinoma	tumour grade, decreasing levels after treatment	serum	[190]
	↑	gastric cancer	tumour grade and metastasis	plasma, serum	[191,192]
	↑	gestational diabetes	body mass index, fasting plasma glucose	plasma	[193]
	↑	glioblastoma multiforme	level of tissue expression, progression	serum	[66,166,194]
	↑	hepatocellular carcinoma	diagnosis	serum	[195]
	↑	laryngeal squamous cell carcinoma	lymph node metastasis	serum	[196]
	↑	multiple myeloma	disease stage	plasma	[197]
	↑	non-small cell lung cancer	histology subtype and tumour-node-metastasis stage	plasma	[198]
	↑	osteoarthritis	diagnosis	plasma	[199]
	↑	papillary thyroid carcinoma	tumour grade	serum	[200]
	↑	rheumatoid arthritis	decreasing levels after treatment	serum	[201]
	↑	systemic lupus erythematosus	level of IL-6	serum	[202]
	↓	acute myocardial infarction	diagnosis	plasma	[203]
	↓	atherosclerosis	diagnosis	plasma	[203]
LINC00467	↓	acute myeloid leukaemia	increasing levels after treatment	serum	[101]
LINC00565	↑	glioblastoma multiforme	survival pattern patients	serum	[73]
LINC00641	↑	Crohn’s disease	diagnosis	serum	[204]
	↑	glioblastoma multiforme	survival pattern patients	serum	[73]
	↑	ulcerative colitis	diagnosis	serum	[204]
MALAT1	↑	acute kidney injury	diagnosis	serum	[205]
	↑	acute pancreatitis	diagnosis	serum	[206]
	↑	angina pectoris	severity of disability	serum	[207]
	↑	breast cancer	diagnosis	serum	[208]
	↑	gastric cancer	metastasis	plasma	[209]
	↑	gestational diabetes	diagnosis	serum	[210]
	↑	glioblastoma multiforme	TMZ chemoresistance	serum	[38]
	↑	hypertension	diagnosis	plasma	[211]
	↑	multiple sclerosis	diagnosis	serum	[212]
	↑	nasopharyngeal carcinoma	tumour stage, decreasing levels after treatment	serum	[213]
	↑	non-Hodgkin lymphoma	tumour stage	plasma	[214]
	↑	non-small cell lung cancer	tumour grade and metastasis	serum	[215]
	↑	osteosarcoma	survival pattern patients	serum	[216]
	↑	ovarian cancer	metastasis	serum	[217]
	↑	Parkinson’s disease	degree of inflammation	serum	[218]
	↑	prostate cancer	diagnosis	serum	[219]
	↑	rheumatoid arthritis	diagnosis	plasma	[220]
	↑	sepsis	clinico-pathological symptoms	plasma	[221]
	↑	severe pneumonia	prediction of survival of patients	serum	[222]
	↑	tongue squamous cell carcinoma	diagnosis	serum, plasma	[223]
	↑	ulcerative colitis	diagnosis	plasma	[224]
	↓	diabetes mellitus 2	diagnosis	serum	[225]
	↓	retinoblastoma	diagnosis	serum	[226]
MIR210HG	↑	glioma	diagnosis	serum	[227]
RPSAP52	↑	renal failure	diagnosis	plasma	[228]
	↓	diabetic retinopathy	diagnosis	plasma	[229]
SAMMSON	↑	glioblastoma multiforme	diagnosis	plasma	[230]
	↑	oral squamous cell carcinoma	levels of tissue expression, decreasing levels after treatment	serum	[231]
	↑	papillary thyroid carcinoma	diagnosis	plasma	[232]
SOX2-OT	↑	head and neck squamous cell carcinoma	diagnosis	plasma	[233]
	↑	lung squamous cell carcinoma	tumour size and lymph node metastasis, decreasing levels after treatment	plasma	[234]
	↑	ovarian cancer	diagnosis	plasma	[235]
	↑	pulmonary arterial hypertension	diagnosis	serum	[236]
TP73-AS1	↑	non-small cell lung cancer	prognosis	serum	[150]
	↑	NK/T-cell lymphoma	diagnosis	blood	[96]
ZEB1-AS1	↑	oesophageal carcinoma	diagnosis	serum	[237]
	↑	prostate cancer	diagnosis	serum	[238]
	↓	amyotrophic lateral sclerosis	diagnosis	blood	[239]
	↓	diabetes mellitus	complications of diabetes (diabetic lung)	plasma	[240]

**Table 3 biomedicines-12-00932-t003:** Suggested lncRNA signatures for the selected diseases according to Table 2.

Disease	Deregulated Levels of lncRNA Biomarkers in Peripheral Blood
breast cancer	ANRIL (↑), HOTAIR (↑), MALAT1 (↑), GAS5 (↓)
coronary artery disease	ANRIL (↑), HOTAIR (↑), GAS5 (↓)
diabetes mellitus 2	ANRIL (↑), HOTAIR (↑), GAS5 (↓), MALAT1 (↓), ZEB1-AS1 (↓)
gastric cancer	HOTAIR (↑), MALAT1 (↑), DGCR5 (↓),
glioblastoma multiforme	ANRIL (↑), HOTAIR (↑), LINC00641 **(**↑)**,** LINC00565 (↑), MALAT1 (↑), SAMMSON (↑), GAS5 (↓)
hepatocellular carcinoma	CRNDE (↑), HOTAIR (↑), CASC2 (↓), DGCR5 (↓), GAS5 (↓)
multiple sclerosis	GAS5 (↑), MALAT1 (↑), ANRIL (↓)
non-small cell lung cancer	ANRIL (↑), CRNDE (↑), GAS5 (↑), HOTAIR (↑), MALAT1 (↑), TP73-AS1 (↑), ADAMTs9-AS2 (↓)
rheumatoid arthritis	HOTAIR (↑), MALAT1 (↑), CASC2 (↓), GAS5 (↓)
sepsis	ANRIL (↑), CRNDE (↑↓), CASC2 (↓), GAS5 (↓)
ulcerative colitis	ANRIL (↑), LINC00641 (↑), MALAT (↑)

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
