# Peer review of "lncRNA Biomarkers of Glioblastoma Multiforme†"

_biomedicines, 2024, doi:10.3390/biomedicines12050932_

Round 1

Reviewer 1 Report

Comments and Suggestions for Authors

The revision article presented by Pokorná et al. titled "LncRNAs as Biomarkers of Glioblastoma Multiforme" provides an interesting and up-to-date overview in the field of oligonucleotide molecules with non-transcriptional activity but with an impact on various cancer functions. A special focus has been made on the implication of Long noncoding RNAs in glioblastoma. My comments are as follows:

  1. The introduction emphasizes that a systematic review was conducted, although the methodology for article search and selection for inclusion in the analysis, as well as the type of analysis used to prepare the review, were not described. It is important to include or clarify this aspect.

  2. In the section "Molecular Mechanisms Underlying lncRNA Functions," various mechanisms of action of lncRNA have been summarized, accompanied by very interesting illustrative diagrams. Why is there no diagram included for the first described mechanism (1. Signals)?

  3. I am confused by the relationship intended between section 4. "LncRNAs and Cancer" and section 5. "LncRNAs and GBM." Could you describe what section 4 aims to contribute to the present review?

  4. In the table provided in section 5, it would be interesting to add the classification of each type of LncRNA according to the mechanisms of action described in section 3.

  5. An illustrative diagram summarizing the different LncRNAs proposed as biomarkers for GBM would help to summarize all the text presented in this section.

Comments on the Quality of English Language
  1. Please review the entire article's wording, as I have found several grammatical errors throughout.

Author Response

Dear Reviewer, 
thank you for your valuable comments and advice. Your comments have improved our article - we have incorporated most of them. Our responses to your comments:

Comment 1: The introduction emphasizes that a systematic review was conducted, although the methodology for article search and selection for inclusion in the analysis, as well as the type of analysis used to prepare the review, were not described. It is important to include or clarify this aspect.
Re: We agree with your opinion, that needed to be added. We have added a paragraph clarifying methodology for article selection, literature review and writing the manuscript (paragraph 2., page 3, line 128). 

Comment 2: In the section "Molecular Mechanisms Underlying lncRNA Functions," various mechanisms of action of lncRNA have been summarized, accompanied by very interesting illustrative diagrams. Why is there no diagram included for the first described mechanism (1. Signals)?
Re: We have added the missing Molecular Mechanism Signals (Figure 1) (page 5, line 193).
Authors appreciate reviewers’ interest in the figures and complimenting remarks. 

Comment 3: I am confused by the relationship intended between section 4. "LncRNAs and Cancer" and section 5. "LncRNAs and GBM." Could you describe what section 4 aims to contribute to the present review? (now section 5 and 6)
Re: We apologise for the confusion. We changed the title of the section to reflect more accurately its content (page 7, line 211). This section presents a general summary of lncRNAs deregulation in multiple cancer types. Also mentioned here is the first established lncRNA detection assay, PCA3 in prostate cancer. This illustrates well the potential for introducing lncRNA detection into practice and the quality of functionality. This section is followed by focused discussion of lncRNA of GMB (page 8, line 250). 

Comment 4: In the table provided in section 5 (6), it would be interesting to add the classification of each type of LncRNA according to the mechanisms of action described in section 3 (4).
Re: We agree with reviewer’s suggestion. However, completing the distribution of lncRNAs according to their molecular mechanisms in the table is unfortunately impossible at present, because for many the molecules in question the exact mechanism of contribution to GMB is not yet known. 

Comment 5: An illustrative diagram summarizing the different LncRNAs proposed as biomarkers for GBM would help to summarize all the text presented in this section.
Re: Thank you for this constructive suggestion. We have created a flowchart for the clinical use of the suggested lncRNA signature (figure 5, page 19, line 211). 

Thank you for your comments and advice. We have re-read the whole article and corrected the errors found in the English.

Reviewer 2 Report

Comments and Suggestions for Authors

The authors submitted a narrative review in which they elucidated a role of long noncoding RNA in detection, risk stratification and prognosis of brain neoplsia. Although the findings seem to be impressive and practically important, I would like to make several comments to discuss them.

1. The authors should concentrate on the tissue specificity and prediction of prognosis. Yet, there is a lack of clear comparisson between conventional score(s) and biomarker models based on the Long noncoding RNA.

2. ALong with it, economical burden of the biomarker implementation might be reflected in the text of the paper.

Author Response

Dear Reviewer, 

Thank you for your valuable advice and comments. Your ideas have inspired us. Our responses to your comments are as follows:

Comment 1: The authors should concentrate on the tissue specificity and prediction of prognosis. Yet, there is a lack of clear comparisson between conventional score(s) and biomarker models based on the Long noncoding RNA.

Re: We have written a review article summarizing lncRNA. In the second part, we focused on the detection of lncRNA in blood for disease prediction, with a focus on GBM, for which there is no diagnostic method from blood yet. We propose the analysis of lncRNA signatures from blood as an early prediction of GBM, and thus faster diagnosis and early treatment. This is not a replacement for existing diagnostic methods. We need further experiments to expand lncRNA signatures to refine prognosis and diagnosis. We hope that our paper will stimulate this discussion and inspire further research on lncRNAs in the blood of GBM patients and as a tool for monitoring treatment success. In brain tissue, detection of lncRNAs could help to better identify the GBM subtype and better target treatment. We believe that our systematic review can be a supportive source of information for further studies to evaluate the establishment of lncRNA as a biomarker for this aggressive tumor. 

Comment 2: Along with it, economic burden of the biomarker implementation might be reflected in the text of the paper.

Re: Estimating the total economic cost is beyond our knowledge and depends on the methods, assays and devices used. Real time PCR is a relatively expensive method, but after the COVID pandemic it is widely available in the health laboratory network. 

Thank you for your contribution to our review article.

Round 2

Reviewer 1 Report

Comments and Suggestions for Authors

The authors have provided additional information and graphics that enhance the understanding of the article. I have no further comments.